# Impact of Varied Recycled Aggregate Inclusions on Mechanical Properties and Damage Evolution Based on Multiphase Inclusion Theory

**DOI:** 10.3390/ma18235430

**Published:** 2025-12-02

**Authors:** Yongsheng Ma, Tiefeng Chen, Xiaojian Gao, Congkai Jin, Qiong Liu

**Affiliations:** 1School of Civil Engineering, Harbin Institute of Technology, Harbin 150090, China; 22b933087@stu.hit.edu.cn (Y.M.); chentf@hit.edu.cn (T.C.); 2School of Environment and Architecture, University of Shanghai for Science and Technology, Shanghai 200093, China; 222271936@st.usst.edu.cn (C.J.); lq612@usst.edu.cn (Q.L.)

**Keywords:** Multiphase Inclusion Theory, model recycled concrete, stress concentration, Finite Element Method

## Abstract

**Highlights:**

**What are the main findings?**
Hard inclusions disperse stress, enhancing the overall stiffness.Soft inclusions consistently exhibit stress concentration.Similar inclusions as the base materials provide stress compatibility

**What are the implications of the main findings?**
Multiphase Inclusion Theory demonstrates predictive capability for stress concentration.It also supports the advanced design and optimization of composite materials

**Abstract:**

This research investigates stress concentration in model recycled concrete using Multiphase Inclusion Theory (MIT). Natural stone, ceramic tiles, glass, red brick, waste concrete, and aerated brick were selected as inclusions in the model recycled concrete matrix. The influence of these inclusions on stress distribution was thoroughly analyzed through theoretical, experimental, and numerical approaches. The results demonstrate that inclusions with varying elastic moduli and Poisson’s ratios induce substantial variations in stress concentration within the matrix. Low-modulus inclusions like aerated brick cause substantial stress concentration, leading to localized failure, whereas high-modulus materials like natural stone and ceramic tile distribute stress more effectively, mitigating concentration effects. Inclusions like red brick and waste concrete, with elastic moduli similar to the matrix, provide better stress compatibility, resulting in a more balanced stress distribution. This study confirms that MIT is a reliable predictor of stress concentration phenomena in materials with high elastic moduli under compression experiments, with theoretical results closely corresponding to experimental and Finite Element Method (FEM) simulations. This validated reliability supports the advanced design and optimization of composite materials in various engineering applications.

## 1. Introduction

The global construction industry’s pressing need for sustainable development has positioned recycled concrete as a promising green building material [1,2,3]. Its production, which incorporates waste concrete and other materials, diverts construction waste from landfills and reduces the consumption of natural resources, thereby offering substantial environmental benefits [4,5,6]. However, the heterogeneous composition of recycled concrete often induces stress concentration, which can adversely affect its mechanical performance and long-term durability [7,8,9]. Therefore, a thorough understanding of stress concentration is crucial for enhancing its practical application.

Initial research on the mechanical properties of recycled concrete primarily focused on the interface surrounding recycled aggregates [1,5,10]. Subsequently, studies have expanded to examine the microstructure and its link to macroscopic mechanical behavior [11,12,13]. For instance, Tang et al. [14] demonstrated that the microstructural characteristics of recycled aggregates largely govern internal stress distribution. Their detailed experimental and numerical simulation analyses provided critical insights into the role of microstructure in stress concentration phenomena, offering recommendations for improving recycled aggregate microstructure. Similarly, Bosque et al. [10] conducted an in-depth investigation into the interfacial transition zone (ITZ) of recycled concrete, revealing that its high porosity and water absorption tendencies lead to the formation of microcracks, thereby weakening interfacial bonding strength and exacerbating stress concentration under load. Their findings indicate that optimizing the surface treatment of recycled aggregates and enhancing the ITZ microstructure can significantly improve the mechanical properties of recycled concrete. This conclusion aligns with the life cycle analysis work of Van den Heede and De Belie [15], who emphasized that applying appropriate modification techniques during production can mitigate stress concentration and extend the service life. Furthermore, alternative materials like red brick, glass, and ceramic tile are increasingly used as aggregates in recycled concrete, and their distinct physical and mechanical properties substantially influence stress distribution. Letelier et al. [16] found that incorporating brick powder as a partial cement replacement reduces CO_2_ emissions, refines the concrete microstructure, mitigates stress concentration, and improves overall performance. Similarly, Thomas et al. [17] reported that ceramic inclusions can improve compressive strength and minimize stress concentration. Although these studies provide valuable insights, a critical knowledge gap remains. A systematic understanding of the mechanisms by which different types of recycled aggregates (e.g., red brick, glass, ceramic tile) influence stress distribution, rooted in their intrinsic mechanical properties rather than solely their microstructure, is lacking. Consequently, predictively optimizing the use of diverse recycled materials remains challenging.

Multiphase Inclusion Theory (MIT), an extension of the classical Eshelby inclusion problem, provides a robust theoretical framework for predicting the effective properties of composite materials containing inhomogeneities of distinct phases. Its core principle involves solving the elastic field of a matrix containing an inclusion with sub-domains of different properties, using techniques such as the equivalent inclusion method. For concrete materials, MIT offers the key advantage of accurately representing complex microstructures, such as aggregates surrounded by a distinct interfacial transition zone. Compared to simpler two-phase models, MIT enables more precise predictions of overall elastic and fracture properties.

This study systematically investigates the influence of different material inclusions on the stress concentration and elastic properties of recycled concrete using MIT. Building upon the aforementioned research findings, this study explores the impact of various inclusions’ microstructures on stress distribution in recycled concrete and validates the application of MIT in this context through experimental and numerical simulations. This work provides a mechanistic understanding of stress concentration in recycled concrete containing various inclusions and offers a theoretical basis for the design of optimized composite materials.

## 2. Stress Distribution Analysis Based on MIT

### 2.1. Overall Mechanical Properties Analysis Based on MIT

Eshelby proposed that if a region within an isotropic elastic solid is retained while the surrounding material is removed, as shown in Figure 1, this region will spontaneously undergo a specific uniform deformation [18,19]. Due to the presence of surrounding material, stress fields will exist both inside and outside of this region. When assuming that the ellipsoidal region in an infinite medium has different mechanical parameters from the rest of the material, this inhomogeneity disturbs the uniformly applied stress field at a distance. This method for addressing elastic isotropic inclusions embedded in an infinite elastic matrix is known as Eshelby’s theory [20], see Appendix A.

In the ellipsoidal region with a prescribed uniform eigenstrain $\varepsilon^{*}$, the total uniform strain can be expressed as a function of the eigenstrain:(1)$$\varepsilon =S:\varepsilon^{*}$$(2)$$\varepsilon^{*}=\left\{\begin{array}{cc} \varepsilon^{*}(x), & \forall x\in \mathrm{CA}\\ 0, & \forall x\in \mathrm{ITZ} \end{array}\right.$$

Here, S is the fourth-order Eshelby tensor, which depends on the material properties and the shape of the inclusion (i.e., the aspect ratio).

Based on Eshelby’s tensor solution mentioned above, a similar dilute approximation result is obtained by approximating the embedding of a single particle in the entire matrix medium.(3)$${\bar{\varepsilon }}^{\left(1\right)}=\varepsilon^{0}+{\tilde{\varepsilon }}^{\left(1\right)},{\bar{\varepsilon }}^{\left(2\right)}=\varepsilon^{0}+{\tilde{\varepsilon }}^{\left(1\right)}+{\tilde{\varepsilon }}^{\left(2\right)}$$

The above formulas encompass the concepts of eigenstrain and equivalent inclusion in the Mori–Tanaka theory.

Therefore, using the Mori–Tanaka method, composite materials with larger inclusion volumes can be calculated.

For the recycled concrete studied in this paper, the Mori–Tanaka method can be used to calculate its basic mechanical properties.

### 2.2. Determining the Internal and External Stress Fields of the Inclusion

As discussed above, using Eshelby’s theory, after applying a uniform load at an infinite distance from the inclusion, the internal stress and strain fields of the ellipsoidal inhomogeneous body depend on its mechanical properties and shape. The equivalent inclusion method can also be used to determine the strain field outside the ellipsoid.

Using Eshelby’s theory, the equivalent stress in all directions for an infinitely thick elliptical cylinder can be obtained. The components of the eigenstress are independent of the coordinates. The tensor is expressed in matrix form as follows:(4)$$\sigma^{*}=N\varepsilon^{*}=\frac{-4\mu m}{(\kappa +1){\left(1+m\right)}^{2}}\left[\begin{array}{ccc} 1+\frac{2}{m} & 1 & 0\\ 1 & 1+2m & 0\\ 0 & 0 & 1 \end{array}\right]\left[\begin{array}{l} \varepsilon_{x}^{*}\\ \varepsilon_{y}^{*}\\ \gamma_{xy}^{*} \end{array}\right]$$

According to Eshelby’s tensor for an elliptical homogeneous inclusion, the total strain components inside the homogeneous inclusion are(5)$$[\varepsilon ]=[S]\left[\varepsilon^{*}\right]=\left[\begin{array}{ccc} S_{1111} & S_{1122} & 0\\ S_{2211} & S_{2222} & 0\\ 0 & 0 & S_{1212} \end{array}\right]\left[\begin{array}{l} \varepsilon_{x}^{*}\\ \varepsilon_{y}^{*}\\ \gamma_{xy}^{*} \end{array}\right]$$

By simplifying the equivalent inclusion method, structural static load analysis is performed using eigenstress as the basic unknown quantity. The eigenstrain is expressed in terms of eigenstress using the invertible matrix N:(6)$$\varepsilon^{*}=N^{-1}\sigma^{*}$$(7)$$\varepsilon =S\varepsilon^{*}=M\sigma^{*}$$(8)$$M=SN^{-1}$$

According to the two-dimensional Hooke’s law,(9)$$e_{ij}=\varepsilon_{ij}-\varepsilon_{ij}^{*}=C_{ijkl}^{-1}\sigma_{kl}$$

For isotropic materials, it is expressed as(10)$$\varepsilon_{ij}-\varepsilon_{ij}^{*}=\left[\sigma_{ij}-\delta_{ij}\sigma_{kl}\nu /(1+v)\right]/2\mu$$(11)$$\left[\begin{array}{c} \varepsilon_{x}-\varepsilon_{x}^{*}\\ \varepsilon_{y}-\varepsilon_{y}^{*}\\ \gamma_{xy}-\gamma_{xy}^{*} \end{array}\right]=\frac{1}{8\mu }\left[\begin{array}{ccc} \kappa +1 & \kappa -3 & 0\\ \kappa -3 & \kappa +1 & 0\\ 0 & 0 & 8 \end{array}\right]\left[\begin{array}{l} \sigma_{x}\\ \sigma_{y}\\ \tau_{xy} \end{array}\right]$$

When all eigenstrains are zero,(12)$$\left[\begin{array}{l} \varepsilon_{x}\\ \varepsilon_{y}\\ \gamma_{xy} \end{array}\right]=\frac{1}{8\mu }\left[\begin{array}{ccc} \kappa +1 & \kappa -3 & 0\\ \kappa -3 & \kappa +1 & 0\\ 0 & 0 & 8 \end{array}\right]\left[\begin{array}{l} \sigma_{x}\\ \sigma_{y}\\ \tau_{xy} \end{array}\right]$$

For a matrix material containing an elliptical inhomogeneity, uniform stress applied to the matrix is $\left[\sigma_{x}^{0},\sigma_{x}^{0},\tau_{xy}^{0}\right]$. By using the equivalent inclusion method, the elastic field is superimposed with the eigenstress in the inclusion, and the resulting elastic material equation for the composite material is(13)$$\left[\begin{array}{c} \varepsilon_{x}^{0}\\ \varepsilon_{y}^{0}\\ \gamma_{xy}^{0} \end{array}\right]=\frac{1}{8\mu_{1}}\left[\begin{array}{ccc} \kappa_{1}+1 & \kappa_{1}-3 & 0\\ \kappa_{1}-3 & \kappa_{1}+1 & 0\\ 0 & 0 & 8 \end{array}\right]\left[\begin{array}{l} \sigma_{x}^{0}\\ \sigma_{y}^{0}\\ \tau_{xy}^{0} \end{array}\right]$$

The relationship between the total strain and the eigenstress is(14)$$\left[\begin{array}{l} \varepsilon_{x}\\ \varepsilon_{y}\\ \gamma_{xy} \end{array}\right]=\frac{-1}{4\mu_{1}}\left[\begin{array}{ccc} m+m\kappa_{1} & 1-\kappa_{1} & 0\\ 1-\kappa_{1} & \frac{\left(1+\kappa_{1}\right)}{m} & 0\\ 0 & 0 & \frac{m\left(m-2+2\kappa_{1}+\kappa_{1}m\right)+\kappa_{1}+1}{2m} \end{array}\right]\left[\begin{array}{l} \sigma_{x}^{*}\\ \sigma_{y}^{*}\\ \tau_{xy}^{*} \end{array}\right]$$

Similarly, by superimposing the internal stress and the perturbation stress field of the composite material, the stress–strain relationship inside the inhomogeneity can be derived using Hooke’s law:(15)$$\left(\varepsilon_{ij}^{0}+\varepsilon_{ij}\right)=C_{ijkl}^{*}-1\left(\sigma_{kl}^{0}+\sigma_{kl}\right)$$

Here, $C_{ijkl}^{*}$ is the elastic tensor of the inclusion.(16)$$\left[\begin{array}{l} \varepsilon_{x}^{0}\\ \varepsilon_{y}^{0}\\ \gamma_{xy}^{0} \end{array}\right]+\left[\begin{array}{l} \varepsilon_{x}\\ \varepsilon_{y}\\ \gamma_{xy} \end{array}\right]=\frac{1}{8\mu_{2}}\left[\begin{array}{ccc} \kappa_{2}+1 & \kappa_{2}-3 & 0\\ \kappa_{2}-3 & \kappa_{2}+1 & 0\\ 0 & 0 & 8 \end{array}\right]\left[\begin{array}{c} \sigma_{x}^{0}+\sigma_{x}^{*}\\ \sigma_{y}^{0}+\sigma_{y}^{*}\\ \tau_{xy}^{0}+\tau_{xy}^{*} \end{array}\right]$$

After simplification, the solution for the eigenstress is obtained as(17)$$\left[\begin{array}{l} \sigma_{x}^{*}\\ \sigma_{y}^{*}\\ \tau_{xy}^{*} \end{array}\right]=\frac{1}{\Gamma }K\left[\begin{array}{l} \sigma_{x}^{0}\\ \sigma_{y}^{0}\\ \tau_{xy}^{0} \end{array}\right]=\frac{1}{\Gamma }\left[\begin{array}{ccc} K_{11} & K_{12} & 0\\ K_{21} & K_{22} & 0\\ 0 & 0 & \Gamma K_{33} \end{array}\right]\left[\begin{array}{l} \sigma_{x}^{0}\\ \sigma_{y}^{0}\\ \tau_{xy}^{0} \end{array}\right]$$

Here, K is a matrix related to material properties and geometric shape parameters. According to the superposition principle, the stress field of the composite material is(18)$$\sigma^{in}=\sigma^{*}+\sigma^{0}$$

The eigenstrain is determined by the internal solution of the homogeneous inclusion:(19)$$\varepsilon^{*}=N^{-1}\sigma^{*}=\frac{\kappa +1}{4\mu }\left[\begin{array}{ccc} -\frac{1}{2}-m & \frac{1}{2} & 0\\ \frac{1}{2} & -\frac{1}{2}-\frac{1}{m} & 0\\ 0 & 0 & \frac{-{\left(1+m\right)}^{2}}{m} \end{array}\right]\frac{1}{\Gamma }K\left[\begin{array}{l} \sigma_{x}^{0}\\ \sigma_{y}^{0}\\ \tau_{xy}^{0} \end{array}\right]$$

For the stress field of the matrix, according to the specific solution proposed for the exterior of an elliptical inhomogeneity in reference [18], for a point h(x,y) outside the elliptical inclusion, an assumed elliptical inclusion is constructed, assuming the elliptical domain satisfies(20)$$\frac{x^{2}}{a^{2}+\lambda }+\frac{y^{2}}{b^{2}+\lambda }=1$$(21)$$\lambda =\frac{1}{2}\left[x^{2}+y^{2}-a^{2}-b^{2}+\sqrt{{\left(x^{2}+y^{2}-a^{2}+b^{2}\right)}^{2}+4\left(a^{2}-b^{2}\right)y^{2}}\right]$$

In reference [18], the external Eshelby tensor for a three-dimensional ellipsoidal inclusion can be obtained using the following equation:
(22)$$\begin{array}{c} T_{ijkl}(x,y,z)=\delta_{ij}\delta_{kl}\left[\frac{v}{1-v}P_{l}^{\left(1\right)}(\lambda )+P_{iK}^{\left(2\right)}(\lambda )\right]+\left(\delta_{ik}\delta_{jl}+\delta_{il}\delta_{jk}\right)\\ \left[\frac{P_{l}^{\left(1\right)}(\lambda )+P_{j}^{\left(1\right)}(\lambda )}{2}+P_{lJ}^{\left(2\right)}(\lambda )\right]+P_{l}^{\left(3\right)}(\lambda )\left(\delta_{ij}n_{k}n_{l}+\delta_{il}n_{j}n_{k}\right)\\ +P_{j}^{\left(3\right)}(\lambda )\left(\delta_{jk}n_{i}n_{l}+\delta_{jl}n_{i}n_{k}\right)+P_{k}^{\left(3\right)}(\lambda )\left(\delta_{kl}n_{i}n_{j}+\delta_{ki}n_{j}n_{l}\right)\\ +\left[P_{ijkl}^{\left(5\right)}(\lambda )+P^{\left(6\right)}(\lambda )\right]n_{i}n_{j}n_{k}n_{l} \end{array}$$


In the external matrix, there is no eigenstrain; the strain in the matrix is purely elastic:(23)$$\varepsilon_{ij}=e_{ij}=T_{ijkl}\varepsilon_{kl}^{*}$$

The specific expression for Eshelby’s tensor at the external point h can be found in reference [18].

For the plane stress problem, the components are obtained through Hooke’s law for the external stress field and are transformed into matrix form:(24)$$\sigma_{ij}=C_{ijkl}\varepsilon_{kl}=C_{ijkl}T_{mnop}\varepsilon_{op}^{*}$$(25)$$\sigma =\frac{4\mu \rho_{a}\rho_{b}}{\kappa +1}\left[\begin{array}{lll} H_{11} & H_{12} & H_{13}\\ H_{21} & H_{22} & H_{23}\\ H_{31} & H_{32} & H_{33} \end{array}\right]\left[\begin{array}{l} \varepsilon_{x}^{*}\\ \varepsilon_{y}^{*}\\ \gamma_{xy}^{*} \end{array}\right]$$

External stress expression is established in the same form as internal stress, with the external stress represented by the perturbation stress. The eigenstrain is expressed in terms of eigenstress,(26)$$\sigma_{ij}=C_{ijkl}\varepsilon_{kl}=C_{ijkl}T_{klmn}N_{mnop}^{-1}\sigma_{oq}^{*}$$(27)$$\sigma =CTN^{-1}\sigma^{*}=\rho_{a}\rho_{b}\left[\begin{array}{lll} R_{11} & R_{12} & R_{13}\\ R_{21} & R_{22} & R_{23}\\ R_{31} & R_{32} & R_{33} \end{array}\right]\sigma^{*}$$

The external stress field is expressed as a function of $\sigma_{0}$. Therefore, similar to the method used for solving the internal problem of the inclusion, the total stress in the matrix is obtained through the superposition principle:(28)$$\sigma^{out}=\sigma +\sigma^{0}$$

## 3. Experimental Study of Model Recycled Concrete

### 3.1. Concept of Model Recycled Concrete

To investigate the impact of residual mortar adhered to natural aggregates on the performance of recycled concrete, Xiao et al. [2,21,22] introduced the concept of model recycled concrete and conducted comprehensive research on its behavior. By strategically placing pre-treated, regularly shaped aggregates at the geometric center of the mortar matrix, they prepared the specimens and subsequently subjected them to Digital Image Correlation (DIC) testing. The test results were further analyzed in conjunction with the intrinsic properties of the materials, providing deeper insights into the effects of aggregate–mortar interaction on the mechanical performance of recycled concrete.

### 3.2. Specimen Preparation and Model Fabrication

Based on theoretical assumptions, the experiment was designed and fabricated model concrete specimens incorporating six distinct materials: natural stone, ceramic tile, glass, red brick, waste concrete, and aerated brick, designated as MS, MT, MG, MB, MC, and MA, respectively. Each group comprised three specimens, resulting in a total of 18 specimens. The specimens were plate-type, with a central cylindrical aggregate embedded within the mortar matrix. The dimensions of the specimens were 20 mm × 100 mm × 100 mm. Cylindrical aggregates were extracted by core drilling from material plates, yielding 18 cylindrical aggregates with a diameter of 12 mm. The thickness of each aggregate was ground to 20 mm, and the aggregate area constituted 1.13% of the entire specimen. The fully cast model concrete specimens are presented in Figure 2.

### 3.3. Testing Method

DIC is a non-contact, non-destructive deformation detection method. It calculates the surface displacement of a specimen by tracking the positional changes in randomly distributed speckle patterns on the specimen’s surface during loading. By further analysis, the strain distribution can be mapped, enabling the identification of both the location and intensity of strain concentration [23,24,25]. Consequently, mechanical performance parameters of the material, such as the effective modulus or Poisson’s ratio, can be determined based on the observed strain over different time intervals. In this experiment, white paint was sprayed onto the surface of the specimens to enhance contrast, followed by the application of speckles for global strain collection. These speckles were distributed uniformly across the entire test area, as depicted in Figure 3.

Here is the revised version with the use of abbreviations for previously mentioned terms: The prepared plate specimens were subjected to compression testing using an electronic universal testing machine. To ensure uniform axial compression and minimize surface stress concentrations, the upper and lower surfaces of the specimens were polished flat, and lubricant was applied to the compression interfaces. Prior to formal loading, each specimen underwent pre-loading cycles, consisting of three repetitions where the load was increased to 5 kN and then reduced to 0 kN, with a 20 s hold at each stage. The pre-loading rate was set at 0.1 mm/min. During the formal loading phase, displacement control was employed, with an initial loading rate of 0.5 mm/min. Once the load reached 5 kN, the loading rate was adjusted to 0.2 mm/min. The loading was terminated when failure of the specimen was observed or when the displacement reached 5 mm. Throughout the compression test, images of the specimen’s entire front surface were captured using an industrial camera. Data collected included the force and displacement from the testing machine, with displacement measured by the movement of the loading beam. Additionally, DIC was utilized to capture displacement and strain information from the specimen’s surface during the loading process.

Using VIC-2D (Version 7) software, the elastic modulus and Poisson’s ratio of the tested specimens were determined via DIC techniques. Based on the force–displacement data acquired during the mechanical performance tests, the elastic portion of the curve was identified, and images corresponding to time nodes in this section were used to calculate the specimen’s stress. More than 300 data points were randomly selected from regions with high confidence. The values of exx and eyy at these points were obtained. By applying Poisson’s ratio formula, data points with negative eyy values were filtered out, and the Poisson’s ratio and elastic modulus were subsequently calculated by correlating the exx values with the corresponding stress at the selected time nodes.

## 4. Results and Discussion

### 4.1. Mechanical Properties

All specimens were compressed to failure during the compression tests, and their mechanical properties are shown in Table 1. The representative load-displacement curves can also be found in Appendix B.

### 4.2. Damage Evolution

The strain cloud map obtained via DIC technology accurately captures the surface deformation of the material. In comparison, stress concentration typically manifests at the boundaries of inclusions, where the differences in mechanical properties are most pronounced, resulting in abrupt changes in the stress field. Moreover, strain distribution is highly sensitive to local geometric variations and defects, enabling the early detection of potential issues within the material, such as the initiation and propagation of microcracks. Consequently, strain cloud maps are employed to analyze the damage evolution patterns in the specimens.

#### 4.2.1. MS

In the MS specimen, the high elastic modulus and strength of the natural stone aggregate provide effective reinforcement to the surrounding mortar, reducing its likelihood of becoming the failure initiation point. During the initial loading phase, stress concentrates on the upper compressed surface, leading to crushing along the narrow edges. Cracks subsequently propagate vertically through the specimen, forming distinct failure channels and ultimately causing structural failure. The crack paths predominantly follow internal defects, such as voids or unmixed regions. As loading continues, shear cracks emerge at the compressed edges, extending diagonally through the specimen and accompanied by significant displacement and audible cracking sounds. Due to the cooperative deformation between the natural stone aggregate and the mortar, strain distribution remains uniform in the early stages of loading, without notable stress concentration. However, as loading progresses, the mortar transitions into the plastic deformation stage, intensifying stress concentration beneath the stone. This connects with internal micro-defects, triggering crack initiation from these concentrated areas and leading to brittle failure. Despite the presence of stress concentration, the failure pattern eventually resembles that of a plate-like component, avoiding both the aggregate and its adjacent regions. The failure is primarily localized in the triangular area directly above the MS, as depicted in Figure 4a. This phenomenon may be attributed to the robust ITZ between the natural stone aggregate and mortar, which prevents it from becoming the weakest point [26]. Therefore, when utilizing natural stone as an aggregate, special attention must be paid to stress concentration effects. Optimizing the shape and arrangement of the aggregate can mitigate these effects, further enhancing the overall performance of the concrete.

#### 4.2.2. MT

For the MT specimen, the strength and elastic modulus of the tile aggregate are comparable to those of natural stone. Under loading, cracks first appear in the upper region of the mortar and quickly propagate along the edges of the observation surface, leading to failure in the triangular area above the aggregate. The cracks primarily follow an “X”-shaped trajectory, forming a significant failure channel and ultimately resulting in the specimen’s overall failure. During crack propagation, the bond between the aggregate and mortar remains intact, preventing it from becoming the weak point. Figure 4b illustrates that, during the initial loading phase, the tile aggregate and mortar interact to form a “V”-shaped strain concentration zone at the aggregate edges, with cracks initially spreading along this concentrated area. As the cracks rapidly expand, the bond between the aggregate and mortar is not compromised; the cracks primarily extend through the mortar, leading to the overall failure of the specimen, as shown in Figure 4b. The tile aggregate’s high elastic modulus and strength result in behavior similar to that of natural stone aggregate. Despite the tight ITZ between the tile aggregate and mortar, with no evident interface failure, cracks predominantly propagate through the weaker regions in the mortar, causing the overall failure. This suggests that, in practical applications, tile aggregate can be effectively used as a substitute for natural stone, providing similar strength and durability to concrete.

#### 4.2.3. MG

In the MG specimen, the high brittleness and low elastic modulus of the glass aggregate, coupled with its smooth surface, result in weak bonding with the surrounding mortar. Under loading, cracks initially form in the mortar around the aggregate and quickly propagate along its edges, leading to the fracture of both the glass aggregate and the mortar. The crack paths predominantly develop along vertical shear planes, accompanied by audible cracking sounds and significant displacement, ultimately culminating in the specimen’s failure. Figure 5a illustrates that stress concentration first emerges on the left side of the specimen during the initial loading phase, causing cracks to extend vertically and damage both the adjacent mortar and the glass aggregate. This suggests that while glass aggregates possess some degree of compressive strength, their low shear resistance leads to failure primarily at the aggregate–mortar interface. To enhance the performance of glass aggregates in concrete, researchers recommend surface treatment or modifying the mortar mix to improve the bond strength between the aggregate and mortar. Additionally, reducing the proportion of glass aggregates can help mitigate early failure induced by low shear resistance, thereby enhancing the overall compressive and shear performance of the structure.

#### 4.2.4. MC

In the MC specimen, the recycled concrete aggregate exhibits high strength, having undergone a prior hardening process that provides substantial compressive resistance. During the initial loading phase, the bond between the aggregate and mortar is strong, mitigating stress concentration to some extent. However, as loading continues, cracks propagate along the aggregate edges and the mortar interface, creating failure channels that ultimately lead to the specimen’s overall failure. Figure 4c illustrates that in the early stages of loading, the aggregate and mortar deform cohesively, with no immediate crack formation due to their robust bond. As the loading increases, random cracks begin to appear on the specimen’s surface, leading to failure initiated by crushing at the lower compressed edge. This behavior may be attributed to initial defects within the aggregate, which are similar to those found in the mortar, creating localized stress concentration points. To enhance the performance of recycled concrete aggregates, researchers recommend pretreatment methods such as grouting repairs or surface coatings to minimize initial defects, thereby improving compressive strength and extending the structural lifespan. The reuse of recycled concrete aggregates not only promotes resource efficiency but also introduces innovative strategies for enhancing structural performance.

#### 4.2.5. MB

In the MB specimen, the porous structure and low strength of the red brick aggregate act as a focal point for stress concentration during the early stages of loading. This concentration leads to crack formation within the aggregate’s pores, which rapidly propagates along the edges and mortar interface, ultimately resulting in the fracture of the red brick and damage to the surrounding mortar. The cracks primarily propagate along the pores and interface defects, creating significant failure channels and culminating in the specimen’s overall failure. Figure 5b illustrates that, during initial loading, strain concentrates in the vertical direction of the red brick aggregate. As the loading progresses, cracks gradually expand through the pores, forming prominent failure channels. Figure 5b further depicts the crack propagation along the pores and interface, ultimately causing the aggregate’s fracture and surrounding mortar damage. To enhance the performance of red brick aggregates in concrete, researchers recommend increasing the aggregate’s strength or using high-strength mortar to minimize initial defects, thereby improving the specimen’s compressive strength and durability. While the porous structure of red brick can offer benefits in certain applications, its low strength renders it unsuitable as a primary aggregate in load-bearing structures.

#### 4.2.6. MA

In the MA specimen, the high porosity, low elastic modulus, and limited compressive strength of the aerated brick aggregate result in pronounced brittleness under loading. Initially, stress concentrates within the aggregate’s internal pores, gradually spreading outward from the center. As loading progresses, cracks rapidly propagate along the aggregate edges and the mortar interface, forming significant failure channels and ultimately leading to the failure of both the aggregate and mortar. The crack path primarily follows the pores and interface defects within the aggregate, culminating in the overall failure of the specimen. Figure 5c illustrates that stress concentration occurs predominantly in the center of the aggregate, with an “X”-shaped crack propagation path. This behavior is attributed to the low elastic modulus and pore defects in the aerated brick, which hinder effective stress transfer within the aggregate. As cracks expand rapidly from the center, they penetrate the aggregate, crushing it into powder and forming a through-failure pattern, as depicted in Figure 5c. This type of brittle failure, accompanied by audible cracking sounds and significant displacement, suggests that aerated brick aggregates cannot effectively resist stress concentration under heavy loads. Due to their low strength and high porosity, aerated bricks are suitable for non-load-bearing structures but pose substantial risks in load-bearing applications. Researchers recommend enhancing the strength of the aggregate or modifying the mortar mix to mitigate stress concentration, thereby improving the overall compressive strength and durability of the concrete structure.

### 4.3. FEM Establishment of Model Recycled Concrete

The model was constructed based on the actual specimen parameters using finite element simulation software to create six types of plate components, each with dimensions of 20 mm × 100 mm × 100 mm, as depicted in Figure 6. To closely replicate the real testing conditions, upper and lower compression plates were included in the model, with a small amount of friction introduced at the contact interfaces to enhance realism. To accurately capture the local characteristics of the stress field and calculate the maximum stress values, a finer mesh was applied to the regions of the aggregate where stress concentration occurs, while a coarser mesh was used in non-stress concentration areas to optimize the total mesh count [27,28]. No scaling factor was applied during the calculations. The material properties were assigned in the model based on previously obtained experimental results for each material phase. Elastic–plastic constitutive parameters were defined for materials that exhibited failure in the experiments, while an elastic constitutive model was used for those that did not experience failure. The loading method mirrored the actual tests, employing a displacement-controlled loading approach. Due to the poor bonding observed between the glass aggregate and mortar during testing, frictional contact was applied in the simulation instead of complete adhesion.

Mechanical performance tests were conducted on several materials used in the production of recycled concrete. Rectangular test specimens, each measuring 50 mm × 20 mm × 100 mm, were cut from six materials: natural stone, glass, red brick, waste concrete, aerated brick, and ceramic tile. In addition, mortar plates measuring 20 mm × 100 mm × 100 mm were cast. The mortar was prepared using P.O.42.5 cement, water, and river sand, with the river sand having a particle size of less than 2.36 mm. The mix ratio of the mortar was mcement:mwater:msand = 1:0.42:3. The compressive strength of the specimens was tested using an electronic universal testing machine, and the elastic modulus and Poisson’s ratio for all seven materials were measured using DIC, as presented in Table 2.

### 4.4. Comparative Analysis of Elastic Stage

Using DIC and finite element software, the elastic modulus of each specimen obtained through different methods can be calculated, as shown in Figure 7.

Figure 7 shows that the results from theoretical calculations, experiments, and numerical simulations are closely aligned, indicating that the model effectively captures the influence of different inclusions on the overall elastic modulus. Figure 7a demonstrates that as the elastic modulus of the aggregate within the mortar decreases, the overall elastic modulus of the specimen correspondingly declines. This occurs because the elastic modulus of the inclusions directly impacts the stiffness of the matrix material.

Hard inclusions, such as natural stone and ceramic tiles, which possess a high elastic modulus, provide greater support to the surrounding mortar under external loading, resulting in a more uniform stress distribution and enhanced overall elastic performance. In contrast, soft inclusions, like aerated brick and waste concrete, have a lower elastic modulus, leading to stress concentration within the matrix during loading, thereby reducing the overall elastic modulus.

During the elastic phase, hard inclusions significantly increase the overall stiffness of the specimen due to their minimal difference in elastic modulus with the mortar, facilitating smoother stress transfer. Numerical simulations indicate that the presence of hard inclusions reduces local stress concentration, thereby enhancing elastic performance in the early stages of compression. However, simulation results tend to be higher than experimental data, as they assume perfect bonding between inclusions and mortar, overlooking the complexity of the microscopic interface.

Soft inclusions, with their lower elastic modulus, fail to adequately support the matrix during loading, leading to significant stress concentration in the surrounding mortar. This reduces the overall stiffness of the specimen, a finding corroborated by both theoretical calculations and experimental results. The deformation of soft inclusions during loading disrupts the stress distribution within the mortar, further diminishing the specimen’s overall stiffness.

Figure 7b shows the average elastic modulus of the experimental, theoretical and simulated results with dots and the error among them with the color range, which indicates that the MR and MRC groups have the lowest error. The comparison between theoretical calculations and experimental results confirms that the theoretical model, which incorporates the elastic properties of inclusions and stress field distribution, accurately reflects the impact of hard and soft inclusions on the overall elastic modulus. Although numerical simulation results are generally higher due to the assumption of perfect bonding, this discrepancy provides valuable insight into the differential effects of various inclusion types on the elastic properties of recycled concrete.

### 4.5. Comparative Analysis of Stress Concentration

To combine with the subsequent experiments, the theoretical analysis uses the model recycled concrete mentioned later as the research target and performs theoretical calculations based on its experimental design. The mortar portion of the model is simplified to a 100 mm × 100 mm square region, with a circular inclusion of 12 mm in diameter at the geometric center of the mortar, as shown in Figure 2.

As mentioned above, the stress and strain distribution inside the circular inclusion is uniform. The internal stress $\sigma^{in}$ will vary depending on the inclusion material but is independent of the location within the inclusion. The stress $\sigma^{out}$ outside the inclusion in the matrix will change with location, so in this paper, $\frac{\sigma^{out}}{\sigma^{in}}$ is set as the stress concentration factor. Figure 8 takes aerated brick aggregate as an example, showing that the sudden changes in the stress concentration factor mainly occur around the circumference of the central aggregate. Figure 8 also shows densely packed iso-stress contours red doted lines near the aggregate, which gradually disperse with increasing distance from it. This is set as the main research object. Therefore, starting from (0, 50) along the positive x-axis direction and around the circular inclusion along the positive y-axis direction, a red path is obtained as shown in Figure 9a, with the endpoint at (100, 50). Along the red line path, only the unidirectional stress in the vertical direction is considered. The stress concentration factors for the six models mentioned above are calculated.

By studying the behavior of these inclusions under stress, as depicted in Figure 9b, we can gain critical insights into the stress fields and concentration phenomena that arise within the material.

(1) Inclusions with high Elastic modulus (natural Stone, ceramic tile and glass)

For inclusions such as natural stone, ceramic tile, and glass—characterized by high elastic moduli relative to the matrix—the stress distribution remains relatively even. These high-modulus materials ensure that applied stress is spread over a larger area of the matrix, resulting in a smoother stress field. As illustrated in Figure 9b, the stress concentration factors for these inclusions remain low throughout the matrix. Numerical results indicate that the high elastic modulus of these inclusions allows them to share the applied load effectively with the surrounding material, minimizing localized stress concentrations.

From a numerical standpoint, Figure 9 demonstrates that the stress concentration factor remains near the baseline, particularly near the boundaries of high-modulus inclusions. This indicates that inclusions like natural stone and ceramic tile are highly efficient in preventing stress concentration. The elastic stiffness of these inclusions balances the internal forces within the concrete matrix, allowing for a near-uniform stress distribution.

The stress concentration factor for high-modulus inclusions is numerically minimal, with values approaching zero in some regions, reflecting effective load transfer between the inclusion and the matrix. The lack of significant stress peaks in the data suggests that these inclusions are well-suited for applications where optimized stress distribution is essential. The data further reveals that as the distance from the inclusion increases, the stress concentration factor gradually decreases, underscoring the inclusions’ role in diffusing stress across the matrix.

(2) Inclusions with Low Elastic Modulus (Aerated Brick)

Inclusions with lower elastic moduli, such as aerated brick and waste concrete, exhibit markedly different stress distribution patterns. Due to their lower stiffness, these materials tend to concentrate stress at the interface between the inclusion and the matrix. As shown in Figure 9b, stress concentration factors for these inclusions are significantly higher compared to those of higher modulus inclusions. This occurs because low-modulus inclusions are less capable of resisting deformation under load, leading to a non-uniform stress distribution.

The numerical data in Figure 9 shows that the stress concentration factor rises sharply at the boundary of low-modulus inclusions. This steep increase suggests that these inclusions cannot effectively transfer stress to the surrounding matrix, leading to localized regions of high stress. While the stress concentration factor decreases with distance from the inclusion, the overall gradient remains steeper compared to high-modulus inclusions.

From a numerical perspective, stress concentration factors for low-modulus inclusions reach significantly higher values, reflecting their inability to evenly distribute applied loads. The peaks in the stress concentration factor near the inclusion boundary, as shown in Figure 9b, indicate that these materials are more prone to inducing stress concentration effects. The greater difference in modulus between the inclusion and matrix leads to more pronounced fluctuations in the stress concentration factor, particularly in regions near the inclusion.

(3) Inclusions with Elastic Modulus Similar to the Matrix (Red Brick, Waste Concrete)

Waste concrete and red brick represent inclusions with elastic moduli similar to that of the matrix. Figure 9b shows that the stress concentration factors for waste concrete and red brick inclusions are moderate, indicating that these inclusions can distribute stress more evenly than low-modulus inclusions like aerated brick, but not as effectively as high-modulus inclusions such as natural stone or ceramic tile. The data suggests that waste concrete and red brick inclusions provide better stress compatibility with the surrounding matrix, resulting in a more balanced stress distribution.

Numerically, stress concentration factors for waste concrete and red brick exhibit smoother profiles compared to both high- and low-modulus inclusions. This behavior suggests that these inclusions integrate well with the matrix, reducing the likelihood of sharp stress gradients. The relatively uniform stress distribution is a result of the similarity in elastic properties between the inclusions and the matrix, promoting effective load sharing across the material.

The stress concentration factors for waste concrete and red brick inclusions in Figure 9 data are moderately low, reflecting a balance between stiffness and load-bearing capacity. The absence of extreme peaks in the data suggests that these inclusions can help reduce overall stress concentration in the concrete matrix. The smoother transitions in the stress concentration factor as the distance from the inclusion increases indicate that waste concrete and red brick inclusions promote more gradual stress redistribution, thereby enhancing the overall structural performance of the recycled concrete.

(4) Analysis of Stress Concentration Peaks and Gradients

One of the critical insights derived from Figure 9’s data is the behavior of stress peaks at the boundaries of inclusions. For low-modulus inclusions, the stress concentration factor reaches its maximum at the interface between the inclusion and the matrix. These peaks indicate areas where the stress field is most disrupted, potentially serving as initiation points for damage. In contrast, high-modulus inclusions exhibit lower stress peaks, as their stiffness helps to mitigate the disruption to the stress field at the inclusion boundary.

Figure 9 also provides valuable insights into the stress gradient, or how the stress concentration factor changes with increasing distance from the inclusion. For high-modulus inclusions, the gradient is relatively shallow, indicating a more uniform stress distribution. Conversely, low-modulus inclusions exhibit steeper gradients, reflecting the rapid decay of stress concentration as distance from the inclusion increases. This sharp gradient in low-modulus inclusions can lead to stress localization, increasing the likelihood of localized failure.

The stress concentration phenomenon observed at the ends of inclusions is closely tied to boundary conditions, as well as the shape and size of the inclusion. In these regions, the interface between the inclusion and the matrix material causes significant local changes in stress. Inclusions with lower elastic moduli produce relatively large displacements during deformation, which redistributes the stress at the ends of the inclusion, resulting in pronounced stress fluctuations. Negative fluctuations in the stress concentration factor suggest that in these regions, the stress experienced by the matrix material is less than the average stress value in the surrounding area. To mitigate these effects, optimizing the shape and distribution of inclusions can reduce stress concentration, thus improving the overall strength and toughness of composite materials.

In regions of high stress concentration, particularly at extremum points of the stress concentration factor and at the fluctuation regions near the inclusion ends, significant damage evolution is likely to occur. The initial stage of damage typically begins with the initiation and propagation of microcracks, which are concentrated in areas where the stress concentration factor is exceptionally high or low. Inclusions with lower elastic moduli, due to their higher deformation capacity, are more prone to microcrack formation under external loads.

As the load increases, these microcracks may propagate and connect, eventually forming macroscopic cracks. The progression of this process is heavily influenced by the stress concentration factor. In the ‘stress shadow zones’ at the ends of the inclusion, where the stress concentration factor is negative, stress is relatively low, and the propagation of microcracks may be slower. However, at the point of maximum stress concentration, microcrack propagation can accelerate significantly.

From a microscopic perspective, the damage evolution process involves the initiation, propagation, and coalescence of microcracks. Initial damage often originates at the interface between the matrix and the inclusion, where stress concentration is most pronounced. Inclusions with lower elastic moduli lead to uneven stress distribution at the interface due to their greater deformation capacity, making it easier for microcracks to form.

As the external load continues to increase, these microcracks may propagate to form a network around the inclusion. As these cracks spread, they may connect within the stress concentration regions at both ends of the inclusion, eventually leading to macroscopic cracks. The speed and path of crack propagation are significantly influenced by the shape and distribution of the inclusion.

The stress concentration factor is a critical variable in the damage evolution process. Higher stress concentration factors lead to faster damage evolution. In regions with extremely high stress concentration factors, the initiation and propagation of microcracks are significantly accelerated, potentially resulting in the rapid formation of macroscopic cracks. To reduce the impact of stress concentration, the shape and distribution of inclusions can be optimized. For instance, designing inclusions with smoother shapes and fewer sharp edges and corners can effectively reduce stress concentration. Additionally, distributing inclusions evenly within the matrix material can help reduce local stress concentrations, slowing the rate of damage evolution.

As shown in Figure 10a–f, the vertical S22 stress cloud maps obtained from finite element simulations were superimposed onto the coordinates in Figure 9 using image processing software. The region with the maximum compressive strain is shown in red, and the area with the minimum compressive strain is depicted in purple. Subsequent analysis was conducted based on these coordinates for detailed comparison.

The coordinates of the maximum strain values in the strain cloud images from the finite element simulation were recorded along the theoretical path around the aggregate circumference. Since the theoretical results include positive and negative extrema, the coordinates at these extrema were normalized. The overall process is as follows. The coordinates of the positive and negative extrema of the stress concentration factor in Figure 9b were processed and listed in Figure 11.

Given that the path is consistent, the y-coordinates correspond to the x-coordinates, so only the x-coordinates were processed. If the midpoint of the x-axis is denoted as M, then the normalized result is,(29)$$x_{M}=\left|\frac{x-M}{50}\right|$$

The data in Figure 11 reveals that theoretical calculations and numerical simulations exhibit distinct stress concentration trends during the elastic phase for each specimen type. The elastic modulus plays a critical role in influencing the overall mechanical performance of the matrix material, as the stiffness and hardness of the inclusions directly govern their interaction with the surrounding matrix. To further investigate how the properties of inclusions impact the overall stress field distribution and failure modes, the analysis can be divided into three categories: hard inclusions, soft inclusions, and inclusions with an elastic modulus similar to those of the matrix. By examining the stress concentration states presented in the table, it is possible to reasonably infer the likely failure patterns for each case.

(1) Hard Inclusions (MS, MT, and MG)

For hard inclusions such as natural stone, ceramic tile, and glass (represented by the MS, MT, and MG specimens), the elastic modulus is significantly higher than that of the mortar matrix. Both numerical simulations and theoretical calculations indicate that these hard inclusions enhance the stiffness of the matrix. As shown in Figure 11, the primary peak values for the MS and MT specimens are both zero, while the secondary peak values are approximately 0.1127, exhibiting a consistent trend in the numerical simulation results. This indicates that hard inclusions provide strong support in the elastic phase, promoting a relatively uniform stress distribution between the inclusion and the matrix during the early stages of loading.

However, Figure 11 also reveals subtle differences in the numerical simulation peak values across different specimens, reflecting the influence of material properties and interface conditions on stress concentration. For the MS and MT specimens, the high elastic modulus of natural stone and tile facilitates effective stress dispersion during the initial stages, resulting in closely matched secondary peak values in both theoretical calculations and simulations. The primary peak value of zero suggests that the highest stress concentration occurs at the center of the inclusion, influenced by the high elastic modulus of the hard inclusion.

In contrast, the MG specimen exhibits slight variations in the numerical simulation results, with a theoretical primary peak value of zero and a secondary peak value of 0.11258, similar to other hard inclusions. However, its simulation peak coordinate is 0.0140, indicating a deviation from other hard inclusions. This discrepancy is primarily attributed to the low adhesion between glass and mortar. The smooth surface of glass inhibits effective bonding, leading to discontinuous stress transfer and localized stress concentration at the interface. This explains the shift in the simulation peak and suggests that under higher loads, interface slippage and cracking are more likely, resulting in failure along the interface rather than at the center of the inclusion.

The comparison between theoretical calculations and numerical simulations demonstrates that the high elastic modulus of hard inclusions significantly impacts stress distribution, with both methods showing similar overall results. However, differences in material properties and interface adhesion can lead to deviations in stress concentration locations in simulations, as observed in the MG specimen due to the low adhesion of glass. The degree and location of the secondary peak shift reflect the complex stress transfer mechanisms between the inclusions and the matrix at various stages of loading.

(2) Soft Inclusions (MA)

Soft inclusions, such as aerated brick (MA specimen), have an elastic modulus significantly lower than that of the matrix. The data in Figure 11 highlight the pronounced stress concentration effect caused by these soft inclusions in concrete specimens. For the MA specimen, the numerical simulation peak coordinate is approximately 0.1120, close to the secondary peak in theoretical calculations. This indicates that soft inclusions are unable to effectively transmit stress during loading, resulting in stress concentration primarily around the inclusion edges. Due to their relatively low load-bearing capacity, these inclusions deform easily under external loads, leading to stress concentration in the surrounding matrix. This results in abrupt local stress changes, initiating damage and crack propagation.

The comparison between theoretical calculations and numerical simulations reveals that the low elastic modulus of soft inclusions makes them incapable of adequately supporting the surrounding matrix during loading. As a result, stress concentration around the inclusion becomes the preferred site for crack initiation and propagation, ultimately leading to brittle failure. In numerical simulations, the peak coordinates suggest that soft inclusions exhibit significant stress concentration early in the loading process, which is characterized by a nonlinear stress–strain relationship. Due to the poor efficiency of stress transfer between the inclusion and the matrix, the edges and interface become critical points for stress concentration, where cracks typically propagate rapidly, forming failure channels. This effect is particularly evident in the MA specimen, where the high porosity of aerated brick further reduces its load-bearing capacity, making it highly susceptible to rapid fracture.

(3) Similar Inclusions to the Matrix (MC and MB)

For inclusions with an elastic modulus similar to that of the matrix, such as waste concrete (MC specimen) and red brick (MB specimen), Figure 11 shows that the numerical simulation peak coordinate for the MC specimen is around 0.1036, slightly lower than the theoretical secondary peak. These inclusions exhibit good synergy with the matrix during the initial loading phase, distributing stress more evenly and suppressing local stress concentration. This is due to their elastic modulus being close to that of the matrix, allowing for effective stress transfer and more uniform stress distribution, which enhances the overall elastic performance.

Theoretical calculations suggest that when the inclusion’s elastic modulus is similar to that of the matrix, it achieves a more harmonious interaction in terms of mechanical properties. Numerical simulations for the MC and MB specimens also indicate that they effectively avoid stress concentration above the inclusion during the elastic phase, demonstrating high compatibility with the matrix. However, as the load increases, new stress concentrations may form at the center of the inclusion or at the inclusion–matrix interface due to deformation. These stress points can gradually lead to crack propagation and eventual failure.

## 5. Conclusions

This study systematically investigated the stress distribution and elastic performance of recycled concrete containing different inclusions. It focuses on the influence of hard, soft, and matrix-similar inclusions on the overall stress field and failure modes. Through comparative analysis of theoretical, experimental, and numerical simulation results, the critical role of the inclusion’s elastic modulus in shaping the mechanical properties of recycled concrete is demonstrated. The key findings are as follows:

(1) The elastic modulus of the inclusions was identified as the pivotal factor governing stress distribution. Hard inclusions (e.g., natural stone and ceramic tile) initially dispersed stress effectively, thereby enhancing the overall stiffness of the composite. However, their high rigidity led to subsequent stress concentration at the inclusion center or interface under increased loading, which became potential sites for crack initiation. In contrast, soft inclusions (e.g., aerated brick) were inefficient for stress transfer, resulting in immediate and pronounced stress concentration around themselves, which significantly weakened structural performance. Inclusions with an elastic modulus similar to the matrix (e.g., red brick and waste concrete) facilitated a more harmonized stress distribution, particularly during the initial loading stage.

(2) The type of inclusion significantly influenced global mechanical behavior and local damage evolution. The numerical simulations showed close agreement with theoretical calculations for hard inclusions, confirming their efficacy in stress sharing within the elastic range. The stress concentration locations predicted by both methods were consistent. A notable exception was glass, where weak interfacial adhesion with the mortar led to a discernible shift in the stress concentration pattern, underscoring the importance of interfacial properties beyond elastic moduli. Conversely, soft inclusions consistently induced significant stress concentration in both experiments and simulations, unequivocally confirming their detrimental effect on load-bearing capacity. While matrix-similar inclusions promoted effective stress dispersion initially, new stress concentration points emerged at their interfaces under higher loads, as corroborated by simulations and theoretical analysis.

(3) The comparison of theoretical, experimental, and numerical simulation results highlights the reliability and accuracy of MIT. MIT demonstrates strong predictive capability for stress concentration in a variety of materials, closely aligning with experimental and simulation data. Notably, for high-modulus materials such as natural stone and tiles, theoretical predictions deviate minimally from actual results, validating MIT’s effectiveness. Although slight deviations occur for low-modulus or brittle materials (e.g., glass, aerated brick) under complex loading, the theory still correctly captures overall stress trends, proving its utility as a robust framework for analyzing composite materials.

(4) This research provides theoretical and practical guidance for designing high-performance composite materials. The integrated approach of combining MIT with numerical simulations enables the prediction of stress concentration from the macro-scale down to the micro-scale, offering valuable insights for microstructural optimization.

Building on these findings, future research should focus on the following directions:

(i) Extending the MIT framework to model inelastic behavior, including crack initiation and propagation, to fully predict the failure process.

(ii) Investigating the behavior of composites with multiple, randomly distributed inclusions of different types to better represent real-world recycled concrete.

(iii) Systematically exploring the effects of inclusion shape, surface texture, and interface treatment on the interfacial strength and overall composite durability.

These investigations will further bridge the gap between theoretical prediction and practical application, enabling the more precise design and optimization of sustainable construction materials.

## Figures and Tables

**Figure 1 materials-18-05430-f001:**
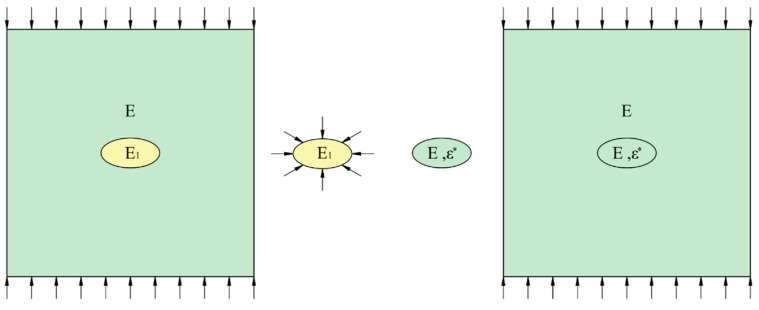
Schematic Diagram of Eshelby’s Method.

**Figure 2 materials-18-05430-f002:**
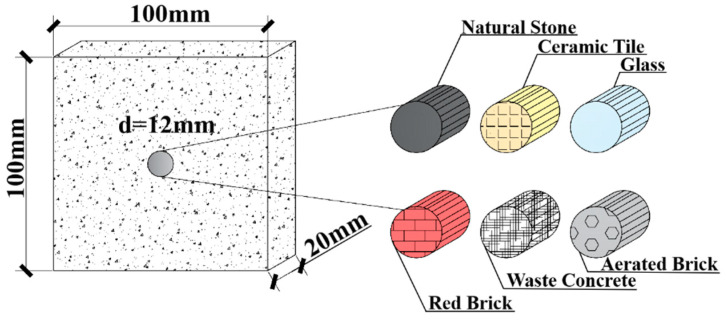
Schematic Diagram of Model Concrete Specimen.

**Figure 3 materials-18-05430-f003:**
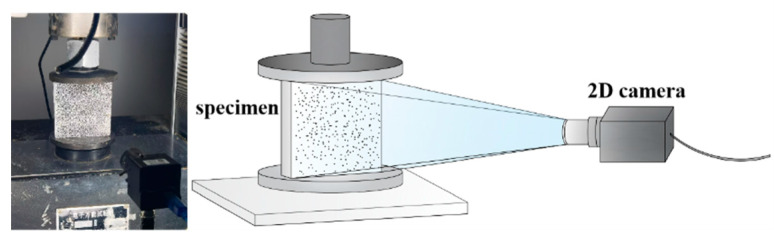
Experimental test set-up.

**Figure 4 materials-18-05430-f004:**
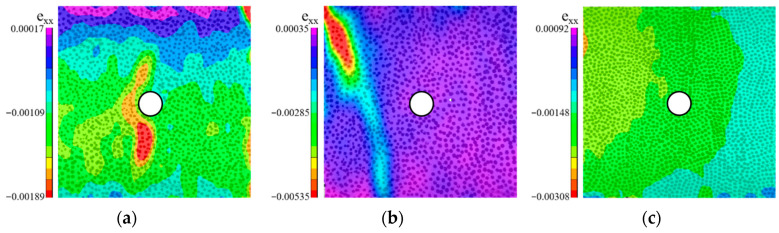
Plate Type Destructive Specimens: (**a**) MS, (**b**) MT, (**c**) MC. Note: The white circle represents the aggregate.

**Figure 5 materials-18-05430-f005:**
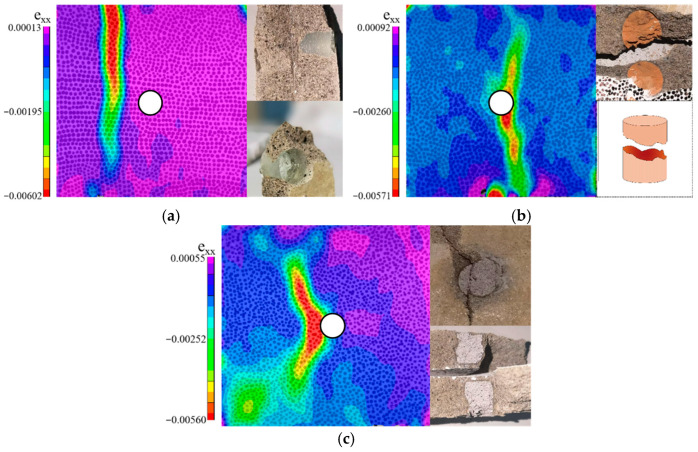
Special Destructive Specimens: (**a**) MG, (**b**) MB, (**c**) MA. Note: The white circle represents the aggregate.

**Figure 6 materials-18-05430-f006:**
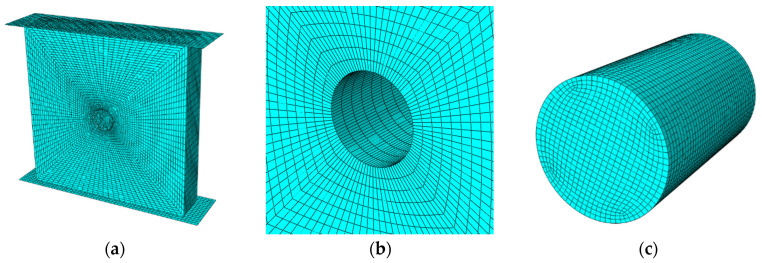
Schematic Diagram of FEM: (**a**) Overall Model, (**b**) Outside mortar, (**c**) Aggregate.

**Figure 7 materials-18-05430-f007:**
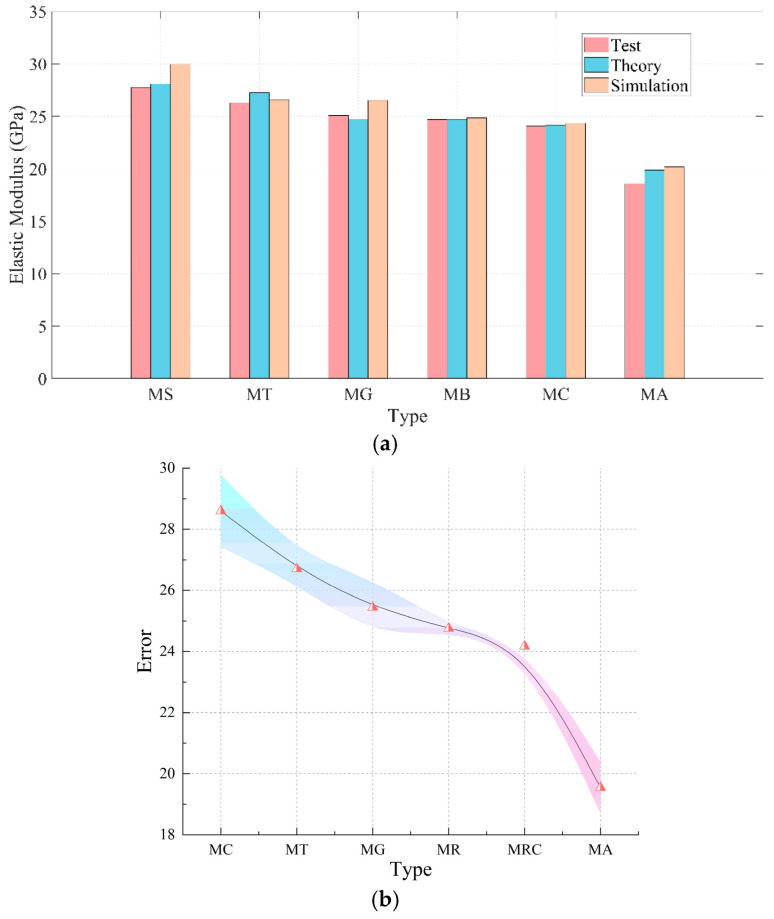
Comparison of elastic modulus: (**a**) Comparison of Elastic Modulus, (**b**) Comparison of Error.

**Figure 8 materials-18-05430-f008:**
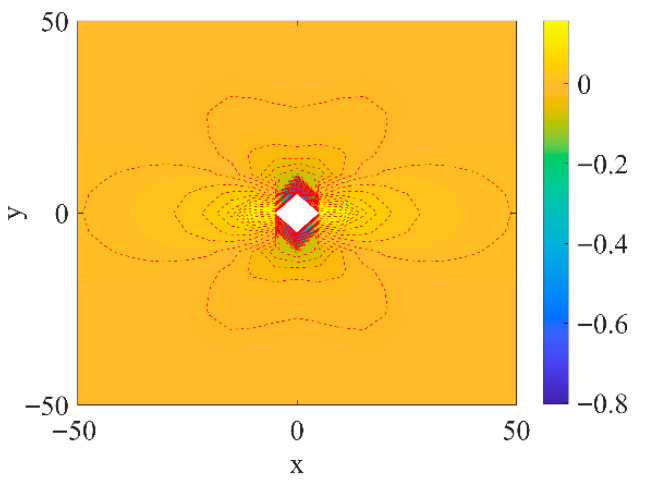
Example Diagram.

**Figure 9 materials-18-05430-f009:**
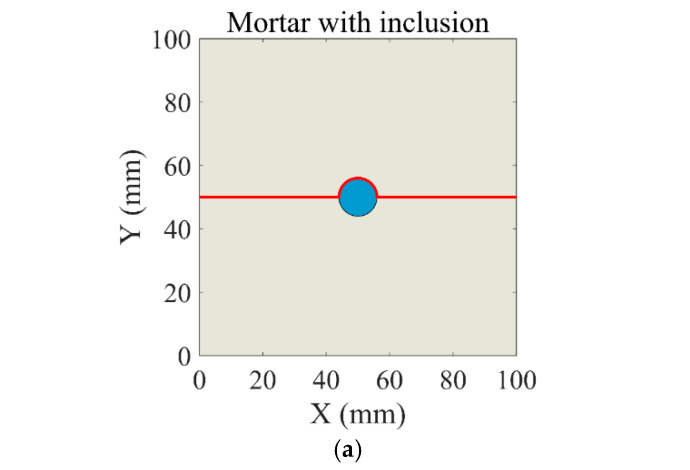
Stress Concentration Factor Diagram: (**a**) Schematic Diagram of Theoretical Model, (**b**) Comparison of Stress Concentration Factors for different Inclusions.

**Figure 10 materials-18-05430-f010:**
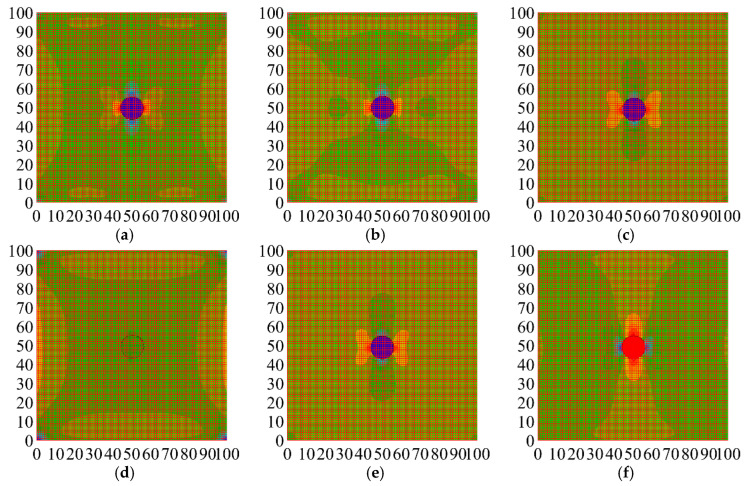
Strain Cloud Image Coordinate Mapping: (**a**) MS, (**b**) MT, (**c**) MG, (**d**) MC, (**e**) MB, (**f**) MA.

**Figure 11 materials-18-05430-f011:**
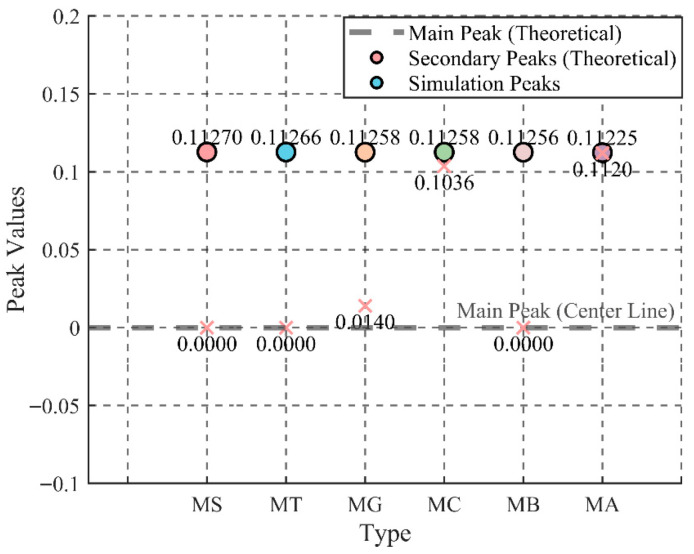
Comparison of Theoretical and Simulated Results.

**Table 1 materials-18-05430-t001:** Mechanical Properties of Model Recycled Concrete.

Type	Elastic Modulus (GPa)	Average (GPa)	Compressive Strength (MPa)	Average (MPa)	Poisson’s Ratio	Average
MS-1	27.65	28.13 ± 0.46	37.17	36.03 ± 1.39	0.2392	0.2396 ± 0.0019
MS-2	28.55		36.54		0.2417	
MS-3	28.25		34.39		0.2380	
MT-1	26.78	27.28 ± 0.47	27.28	28.52 ± 1.25	0.2375	0.2298 ± 0.0091
MT-2	27.68		29.78		0.2322	
MT-3	27.38		28.76		0.2196	
MG-1	23.72	24.22 ± 0.47	24.40	24.14 ± 1.08	0.2285	0.2255 ± 0.0052
MG-2	24.62		22.95		0.2195	
MG-3	24.32		25.08		0.2285	
MB-1	23.86	24.36 ± 0.45	22.93	21.45 ± 1.30	0.2426	0.2391 ± 0.0080
MB-2	24.76		21.10		0.2448	
MB-3	24.46		20.34		0.2299	
MC-1	23.54	24.04 ± 0.46	26.80	26.50 ± 0.52	0.2535	0.2412 ± 0.0115
MC-2	24.44		26.86		0.2312	
MC-3	24.14		25.83		0.2390	
MA-1	18.79	19.29 ± 0.46	18.15	19.63 ± 1.34	0.2195	0.2249 ± 0.0075
MA-2	19.69		19.90		0.2209	
MA-3	19.39		20.83		0.2342	

**Table 2 materials-18-05430-t002:** Physical Properties of Materials.

Material Type	Elastic Modulus (GPa)	Compressive Strength (MPa)	Poisson’s Ratio
Natural Stone	75.0	170.1	0.360
Ceramic Tile	60.0	107.1	0.225
Glass	29.8	97.3	0.221
Red Brick	26.8	9.7	0.217
Waste Concrete	25.0	30.1	0.225
Aerated Brick	2.0	5.0	0.201
Mortar	24.0	33.32	0.238

## Data Availability

The original contributions presented in this study are included in this article. Further inquiries can be directed to the corresponding author.

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
