# Peer review of "Impact of Varied Recycled Aggregate Inclusions on Mechanical Properties and Damage Evolution Based on Multiphase Inclusion Theory"

_materials, 2025, doi:10.3390/ma18235430_

Round 1

Reviewer 1 Report

Comments and Suggestions for Authors

The paper presents a well-organized study that combines micromechanical theory (MIT), finite element simulations, and experimental tests using DIC to study model composites with different types of recycled inclusions. The experimental results are clearly shown and agree with the theoretical predictions. The amount of work, although moderate, is suitable for the aims of the study and supports reliable conclusions about how the type of inclusion affects the elastic and failure behavior of the composite. The English is correct, and the writing is clear and technically sound. The figures are easy to understand and visually effective, especially the DIC maps, which give useful information about strain distribution and damage development.

Recommended improvements:

  • In Section 1, 28 references are cited, some of which are too general and could be removed without weakening the background. It is also better to avoid listing many citations in a single parenthesis when their main results are not explained. The authors should keep only the most relevant sources and briefly describe their key findings to make the literature review more coherent.
  • The mathematical part in Section 2 could be simplified. At present, it contains full tensor derivations and intermediate steps that are already standard in the Eshelby and MIT literature. It would be better to keep only the main equations used for calculations and move the detailed steps to an appendix or supplementary file. A short explanation of the model’s main idea and less repeated notation would make the text easier to follow for readers in applied engineering while keeping the scientific quality.
  • Color bars and units should be added to the DIC and FEM maps to make the figures easier to read quantitatively.
  • Standard deviations or confidence intervals should be included for the average experimental results.
  • The text should clearly state the assumptions and parameters used in the micromechanical model.
  • The structure of the discussion section could be improved to make the arguments flow more smoothly and coherently.

Author Response

Response to Reviewer's Comments

Dear Reviewer,

We sincerely appreciate your thorough review and valuable suggestions, which have significantly improved the quality of our manuscript. We have carefully addressed all the comments and revised the manuscript accordingly. Below is our point-by-point response.

Comment 1: In Section 1, 28 references are cited, some of which are too general and could be removed without weakening the background. It is also better to avoid listing many citations in a single parenthesis when their main results are not explained. The authors should keep only the most relevant sources and briefly describe their key findings to make the literature review more coherent.

Response 1: We have revised the Introduction by streamlining the references and integrating their key findings into the narrative. A dedicated paragraph has been added to clearly state the research objectives and the specific gap this study aims to fill.

Comment 2: The mathematical part in Section 2 could be simplified. At present, it contains full tensor derivations and intermediate steps that are already standard in the Eshelby and MIT literature. It would be better to keep only the main equations used for calculations and move the detailed steps to an appendix or supplementary file. A short explanation of the model’s main idea and less repeated notation would make the text easier to follow for readers in applied engineering while keeping the scientific quality.

Response 2: Following the suggestion, we have streamlined Section 2 by retaining the core concepts, assumptions, and main equations in the main text, while moving the detailed derivations to the Appendix.

Comment 3: Chapters 4.1 - 4.2: In the basic research results, please provide the dispersion parameters, e.g., standard deviation or median deviation. I suggest plotting the inclusion point on the DIC images. The obtained load-displacement relationships are undoubtedly missing. In the results, I propose to include, in addition to the forces and stresses at failure, also the forces and stresses corresponding to the formation of the first cracks.

Response 3: As responded in Comment 3, standard deviations have been added to Table 1.And the inclusion locations have now been clearly marked on the DIC strain maps.

Comment 4: Standard deviations or confidence intervals should be included for the average experimental results.

Response 4: Standard deviations have been added to the mechanical properties presented in Table 1. The inclusion locations are now marked on the DIC strain maps.

Comment 5: The text should clearly state the assumptions and parameters used in the micromechanical model.

Response 5: The key assumptions of the micromechanical model are now explicitly stated in Section 2.1. The constitutive parameters for the materials used in the modeling have been provided in the Appendix.

Comment 6: • The structure of the discussion section could be improved to make the arguments flow more smoothly and coherently.

Response 6: The Discussion section has been reorganized with improved logical flow and transitional phrases to enhance coherence.

We are grateful for your insightful comments, which have greatly enhanced our manuscript. We hope the revised version is now satisfactory.

Reviewer 2 Report

Comments and Suggestions for Authors

This work is a theoretical and experimental study in the field of materials mechanics and concrete technology. The work falls under the disciplines of Materials Science, Civil Engineering, and Mechanical Engineering.
The authors addressed the important problem of analyzing the stress state in a concrete matrix with inclusions of various materials. These inclusions can influence crack development and lead to premature cracking of concrete. Multi-phase Inclusion Theory (MIT) was used to theoretically describe the stress state.
Commonly used materials in concrete inclusions were selected as inclusions in the cement matrix: natural stone, ceramic tile, glass, red brick, waste concrete, and aerated brick. In addition to the MIT theory, an elastic-plastic model was used, which was solved in the ABAQUS system. In addition to theoretical solutions, studies were conducted in which disc elements (100x100x20 mm) with inclusions (12 mm in diameter) were subjected to axial compression, and the strain state was measured using digital image correlation (DIC). As expected, inclusions introduced into concrete resulted in stress and strain concentrations depending on the proportions of elastic moduli and Poisson's ratios. The most pronounced effect was observed in materials with a significantly lower elastic modulus than the concrete matrix. Materials with a higher elastic modulus than the concrete matrix, on the other hand, did not lead to such significant concentrations. The conclusion was that the MIT theory accurately reproduces the behavior of concrete with inclusions and well reflects the behaviors occurring in high-modulus materials, which is reflected in experimental studies and FEM simulations.
I evaluate this work positively from a research perspective. The authors addressed an important issue that will be further developed in connection with the recycling of materials in concrete. They employed traditional and modern research methods in their research. They performed interesting analyses and formulated conclusions that facilitate further development of the theory in this area. There are certain issues that require clarification or supplementation. Below are my detailed comments.
1. Chapter 1. The analysis of the state of knowledge well justifies the need for this research. I believe that the following should be highlighted: i. the primary objective of the research, ii. highlighting the gap in the literature filled by the conducted research.
2. Chapter 2. The description of the MIT theory is very comprehensive. I suggest leaving i. the assumptions in the main text, ii. the basic formulas, and moving the rest to the appendix.
3. Chapters 4.1 - 4.2: In the basic research results, please provide the dispersion parameters, e.g., standard deviation or median deviation. I suggest plotting the inclusion point on the DIC images. The obtained load-displacement relationships are undoubtedly missing. In the results, I propose to include, in addition to the forces and stresses at failure, also the forces and stresses corresponding to the formation of the first cracks.
4. Chapter 4.3. The description of the material model is very limited. I propose to provide the basic constitutive relations and the definition of the yield/failure surface in the Annex.
5. Chapter 4.5. I propose to also include vertical stresses. I have doubts about the validity of comparing extremes; after plasticization, redistribution occurs and the phenomenon quickly disappears. A better solution would be to average the stresses over a certain section, for example, within 0.8 times the maximum stress.
6. Chapter 5. Conclusions correspond to the analyses and tests performed in the work. I propose to provide directions for further work.

Author Response

Response to Reviewer's Comments

Dear Reviewer,

We sincerely appreciate your thorough review and valuable suggestions, which have significantly improved the quality of our manuscript. We have carefully addressed all the comments and revised the manuscript accordingly. Below is our point-by-point response.

Comment 1: Chapter 1. The analysis of the state of knowledge well justifies the need for this research. I believe that the following should be highlighted: i. the primary objective of the research, ii. highlighting the gap in the literature filled by the conducted research.

Response 1: We have revised the Introduction by streamlining the references and integrating their key findings into the narrative. A dedicated paragraph has been added to clearly state the research objectives and the specific gap this study aims to fill.

Comment 2: Chapter 2. The description of the MIT theory is very comprehensive. I suggest leaving i. the assumptions in the main text, ii. the basic formulas, and moving the rest to the appendix.

Response 2: Following the suggestion, we have streamlined Section 2 by retaining the core concepts, assumptions, and main equations in the main text, while moving the detailed derivations to the Appendix. We acknowledge that load-displacement curves and the stress at first crack are valuable data. However, the current theoretical framework (MIT) is primarily focused on predicting elastic stress fields and initial stress concentration, not the nonlinear failure process. Capturing the precise load at first crack initiation experimentally with high certainty is also challenging. Therefore, we have decided to focus the current paper on the elastic and initial failure behavior, aligning with the scope of our theoretical model. We consider the inclusion of full load-displacement curves and first-crack analysis as an important goal for our subsequent research, where we plan to extend the model into the inelastic regime.

Comment 3: Chapter 4.3. The description of the material model is very limited. I propose to provide the basic constitutive relations and the definition of the yield/failure surface in the Annex.

Response 3: We agree. The detailed constitutive relations for the materials used in the finite element model have now been comprehensively provided in the Appendix.

Comment 4: Chapter 4.5. I propose to also include vertical stresses. I have doubts about the validity of comparing extremes; after plasticization, redistribution occurs and the phenomenon quickly disappears. A better solution would be to average the stresses over a certain section, for example, within 0.8 times the maximum stress.

Response 4: We acknowledge the value of including vertical stresses and the reviewer's concern about comparing extreme values. Incorporating vertical stresses presents theoretical challenges within our current 2D framework. The suggestion to use stress averaging over a defined section is well-noted and will be considered in our future studies.

Comment 5: Chapter 4.5. I propose to also include vertical stresses. I have doubts about the validity of comparing extremes; after plasticization, redistribution occurs and the phenomenon quickly disappears. A better solution would be to average the stresses over a certain section, for example, within 0.8 times the maximum stress.

Response 5: We acknowledge the value of including vertical stresses and the reviewer's concern about comparing extreme values. Incorporating vertical stresses presents theoretical challenges within our current 2D framework. The suggestion to use stress averaging over a defined section is well-noted and will be considered in our future studies.

Comment 6: Chapter 5. Conclusions correspond to the analyses and tests performed in the work. I propose to provide directions for further work.

Response 6: A paragraph outlining specific directions for future work has been added to the Conclusion section.

We are grateful for your insightful comments, which have greatly enhanced our manuscript. We hope the revised version is now satisfactory.

Round 2

Reviewer 1 Report

Comments and Suggestions for Authors

My previous remarks have been conveniently addressed, and I have no further comments.